



# Development of a High-Resolution Coupled SHiELD-MOM6 Model. Part I – Model Overview, Coupling Technique, and Validation in a Regional Setup

Joseph Mouallem[1], Kun Gao[1], Brandon G. Reichl[2], Lauren Chilutti[2], Lucas Harris[2], Rusty Benson[2], Niki Zadeh[2], Jing Chen[1], Jan-Huey Chen[2], and Cheng Zhang[1,3]

[1]Program in Atmospheric and Oceanic Sciences, Princeton University, Princeton, NJ, USA
[2]Geophysical Fluid Dynamics Laboratory/NOAA, Princeton, NJ, USA
[3]Department of Civil and Environmental Engineering, Rowan University, Glassboro, NJ, USA

**Correspondence:** J. Mouallem (mouallem@princeton.edu)

**Abstract.** We present a new high-resolution coupled atmosphere-ocean model, SHiELD-MOM6, which integrates GFDL's advanced atmospheric model, System for High-resolution modeling for Earth-to-Local Domain (SHiELD), the Modular Ocean Model version 6 (MOM6), and the Sea Ice Simulator (SIS2). The model leverages the Flexible Modeling System (FMS) coupler and its innovative exchange grid to enable a robust and scalable two-way interaction between the atmosphere and ocean. The atmospheric component is built on the non-hydrostatic Finite-Volume Cubed-Sphere Dynamical Core (FV3) with the latest version of SHiELD physics parametrization suite, while the ocean component is the latest version of MOM supporting kilometer-scale high-resolution and regional applications. Validation of this new coupled model is demonstrated through a suite of experiments, including idealized hurricane simulations and a realistic North Atlantic case study featuring Hurricane Helene 2024. The results reveal that air-sea interactions are effectively captured, by analyzing the storm intensity, structure and its effects on the ocean phenomena such as the upwelling and sea level changes. Scalability tests further confirms the model's computational efficiency. This work established a unified modular cornerstone for advancing high-resolution coupled modeling with significant implication for weather forecasting and climate research.

## 1 Introduction

With the increasingly larger computing resources, advancing weather and climate models to higher resolutions has become both feasible and a natural evolution in model development. This progression enables a more accurate representation of key processes from the explicit simulation of small-scale convection, which often interacts with larger mesoscale atmospheric dynamics, to essential air-sea exchanges and oceanic processes. These improvements are vital for enhancing forecast skill and refining climate projections. Moreover, accurately simulating weather and climate extremes requires models that capture the full spectrum of underlying physical mechanisms, deepening our understanding of event formation, intensity, and impacts. Such capabilities are essential for comprehensive risk assessment and the development of effective mitigation strategies.





Building on decades of advancements in numerical modeling at the Geophysical Fluid Dynamics Laboratory (GFDL), this work introduces the SHiELD-MOM6 model, a high-resolution, coupled system that seamlessly integrates GFDL's advanced atmospheric model, SHiELD, with the Modular Ocean Model version 6 (MOM6) and the Sea Ice Simulator (SIS2). By coupling these components, the model provides a robust framework for investigating air-sea interactions at fine scales, offering new

insights into extreme weather events and climate variability by exploring key processes such as tropical cyclone dynamics, storm surges, oceanic heat transport, and sea ice variability.

Coupling the atmospheric and oceanic components presents inherent challenges due to their differing grid geometries, numerical schemes, and time-stepping protocols. To overcome these obstacles, the SHiELD-MOM6 model employs the Flexible Modeling System (FMS) full coupler along with its innovative exchange grid framework (Balaji et al., 2006). This approach fa-

cilitates robust two-way transfer of prognostic variables, fluxes, and key physical processes between the grid of the atmospheric model and that of the ocean model. In doing so, the framework preserves the conservation of mass, heat, and momentum across scales while accurately capturing the feedback mechanisms critical to any physical phenomenon. This coupling technique has been used in GFDL's flagship models such as SPEAR, CM4 and ESM4 (Delworth et al., 2020; Held et al., 2019; Dunne et al., 2020) and their predecessors for many years, demonstrating its reliability in simulating complex earth system processes across

scales.

This paper is organized as follow. Section 2 describes the primary components of the model, including detailed overviews of the infrastructure layer and the atmosphere and ocean components. Section 3 outlines the coupling methodology including the science framework and software infrastructure, emphasizing the role of the FMS full coupler and exchange grid in coupling the model components and detailing the fluxes exhanged between atmosphere and ocean. Section 4 presents the validation

experiments, discussing both idealized hurricane simulation and a realistic simulation of hurricane Helene of September 2024. Section 5 details the model scalability and computational performance. Section 6 summarizes the key finding and outlines current and future development efforts.

## 2 Model Primary Components

In this section, we present the main components used to build the SHiELD-MOM6 model. GFDL's infrastructure layer the

Flexible Modeling System (FMS) is presented first. Second, the atmosphere model SHiELD based on the finite volume cubed sphere dynamical core (FV3). And last GFDL's Modular Ocean Model version 6 (MOM6) and Sea Ice Simulation version 2 (SIS2). References for each component can be found in the model documentation or through the links in Appendix A

### 2.1 Infrastructure layer

The Flexible Modeling System (FMS) was one of the first modeling frameworks developed to facilitate the construction of

coupled models and has been under continued development since 1998 at GFDL. It is a software environment that supports the efficient development, construction, execution, and scientific interpretation of atmospheric, oceanic, and climate system models written in Fortran for HPC systems. This framework allows an efficient development of numerical algorithms and





computational tools across various high-end computing architectures using common, user-friendly representations of the underlying platforms. It supports distributed and shared memory systems, as well as high-performance architectures. At GFDL, scientific groups can simultaneously develop new physics and algorithms, coordinating periodically through this framework. FMS does not determine model configurations, parameter settings, or choose among various options, as these require scientific research. The development of new component models is also outside its direct scope but supported by infrastructural changes within FMS. The collaborative software review process for contributed models is crucial to FMS. FMS includes:

- Message Passing Interface (MPI) Domain Decomposition: Provides software infrastructure for seamless and efficient utilization of MPI libraries for scalable parallel computations.

- Software Infrastructure: Provides tools for parallelization, I/O, data exchange between model grids, time stepping orchestration, makefiles, and sample run scripts, insulating users from machine-specific details.

- Standardized Interfaces: Ensures standardized interfaces between component models, coordinates diagnostic calculations, and prepares input data. Includes common preprocessing and post-processing software when necessary.

## 2.2 Atmosphere components

The atmospheric component model we use in the regional coupled system is the System for High-resolution modeling for Earth-to-Local Domain (SHiELD), which was built and has been continuously developed at GFDL as an advanced model for a broad range of applications (Harris et al., 2020).

SHiELD employs FV3, a nonhydrostatic finite-volume cubed-sphere dynamic core that has been in development at GFDL for almost three decades (Lin and Rood, 1996, 1997; Lin, 2004; Putman and Lin, 2007; Harris and Lin, 2013; Chen et al., 2013; Harris et al., 2016; Mouallem et al., 2022, 2023). It is used in many weather and climate models for a wide range of applications from short-term weather forecasts to centuries-long climate simulations, moving-nest hurricane forecasts, chemical and aerosol transport modeling, cloud-resolving modeling, and so on (Cheng et al., 2024; Ramstrom et al., 2024; Harris et al., 2023; Bolot et al., 2023; Merlis et al., 2024a, b). FV3 solves the hydrostatic or non-hydrostatic compressible Euler equations on a gnomonic cubed-sphere grid with a Lagrangian vertical coordinate. The algorithm is fully explicit except for fast vertically-propagating sound and gravity waves which are solved by the semi-implicit method. The long time step of the solver also serves as the physics time step. Within each long time step, the user can specify the number of vertical remapping loops, during which subcycled tracer advection is performed. Additionally, the number of acoustic time steps per remapping loop can be set, defining an acoustic time step in which sound and gravity wave processes are advanced, and thermodynamic variables are advected. Coupling with other components, such as the ocean, will occur at intervals corresponding to a multiple of the long timestep (physics timestep).

The detailed description of the solver's horizontal and vertically Lagrangian discretizations can be found in Lin and Rood (1996, 1997) and Lin (2004). FV3's numerics are extensively described in the aforementioned references and will not be repeated here. However, its versatility and computational efficiency make it a strong foundation for a variety of atmospheric modeling applications, including high-resolution weather forecasting and climate simulations.



The physics parameterizations in SHiELD were originally adopted from the Global Forecast System (GFS) physics package but have been heavily updated. Currently, we use the GFDL microphysics scheme (Zhou et al. (2019)), the Eddy-Diffusivity Mass-Flux (EDMF) boundary layer scheme Zhang et al. (2015), the scale aware Simplified Arakawa-Schubert (SAS) of Han et al. (2017), the Noah Land Surface Model of Ek et al. (2003) or Noah-MP of Niu et al. (2011) and a modified version of the

Mixed Layer Ocean of Pollard et al. (1973). Three major SHiELD configurations are being heavily tested and updated continuously: (a) Global SHiELD (Harris et al., 2020; Zhou et al., 2024); (b) T-SHiELD (Gao et al., 2021, 2023); (c) C-SHiELD (Harris et al., 2019; Kaltenbaugh et al., 2022). SHiELD could also be configured differently depending on the application of interest, e.g. S-SHiELD for seasonal to sub-seasonal prediction is being developed. Notably, all SHiELD configurations share the same codebase, executable, and pre/post-processing tools, adhering to the unified modeling philosophy: "one code, one

executable, one workflow." In this work, we employ a regional configuration of SHiELD based on the limited-area configuration of FV3 (Black et al., 2021), which has been widely utilized in both research and operational settings. This setup has demonstrated skill in providing accurate forecasts up to 60 hours with minimal computational resources (Black et al., 2021).

## 2.3 Ocean components

The Modular Ocean Model version 6 (MOM6) and the Sea Ice Simulator version 2 (SIS2), developed at GFDL, provide a robust

framework for simulating ocean and sea ice processes with high accuracy and computational efficiency (Adcroft et al., 2019). MOM6 employs a finite-volume approach on a C-grid, enabling conservation of mass, heat, and tracers while allowing for flexibility in resolving complex oceanic features, such as boundary currents, mesoscale eddies, and thermohaline circulations. Its vertical Lagrangian remapping algorithm allows the usage of any vertical coordinate to remap horizontal layer to a Eulerian reference, implicitly resolving advection and effectively eliminating the Courant–Friedrichs–Lewy (CFL) restriction in the

vertical direction (similar to FV3) (see Griffies et al., 2020). MOM6 is highly configurable, supporting applications ranging from idealized studies to high-resolution global simulations and earth system models.

SIS2 complements MOM6 by simulating the dynamics and thermodynamics of sea ice, including ice growth, melt, deformation, and ridging processes. It incorporates advanced parameterizations to model sea ice interactions with the ocean and atmosphere, such as brine rejection, surface albedo, and momentum fluxes. These capabilities allow SIS2 to capture the

essential feedback mechanisms between sea ice, ocean circulation, and atmospheric forcing.

The coupling of MOM6 and SIS2 through the FMS framework enables the seamless integration of ocean and sea ice dynamics with atmospheric processes. The exchange grid facilitates conservative and accurate flux exchanges, ensuring realistic representation of interfacial processes, such as heat and momentum transfer. The inclusion of SIS2 enhances the model's ability to simulate polar and high-latitude phenomena, such as sea ice extent variability and its impact.

The physical model configuration of MOM6 follows closely to that in Adcroft et al. (2019). We make a few changes, shifting from a hybrid vertical coordinate designed for climate simulation to employ a telescoping $z*$ vertical coordinate that ensures relatively fine grid-spacing is maintained in the upper ocean. The surface grid spacing is is 2m in this configuration, increasing to 10m at about 100m depth. The ocean initial condition imposes a 20 m mixed layer depth everywhere, with a 31°C mixed layer temperature and a gradient of 0.05°C m$^{-1}$ below. Vertical mixing in the ocean surface boundary layer is described by





Reichl and Hallberg (2018), including a wind-speed dependent Langmuir turbulence parameterization following Reichl and Li (2019); Li et al. (2017). Stratified shear-driven mixing is parameterized following Jackson et al. (2008). While MOM6 can be configured to work with open boundary conditions, we do not employ that since it has little impact on the simulations presented here.

## 3   Coupling methodology

This section outlines the coupling methodology. The first subsection thoroughly examines the variables exchanged that underpin the primary physical processes essential to the coupling procedure. The second subsection details the technical framework and implementation strategy that facilitates an efficient exchange and interaction among model components. It is worth mentioning that the land component is still coupled through SHiELD physics suite rather than at the full coupler level. Work is currently in progress to integrate GFDL's latest land model at the coupler level.

**3.1   Physical processes**

As discussed in the previous section, any variable or parameter can be projected between the native grids of model components and the exchange grid. In the current model, several dynamic and physical variables from both the atmosphere and ocean are mapped onto the exchange grid (called Xgrid thereafter) where relevant quantities are computed and then projected back to each component as shown in figure 1.




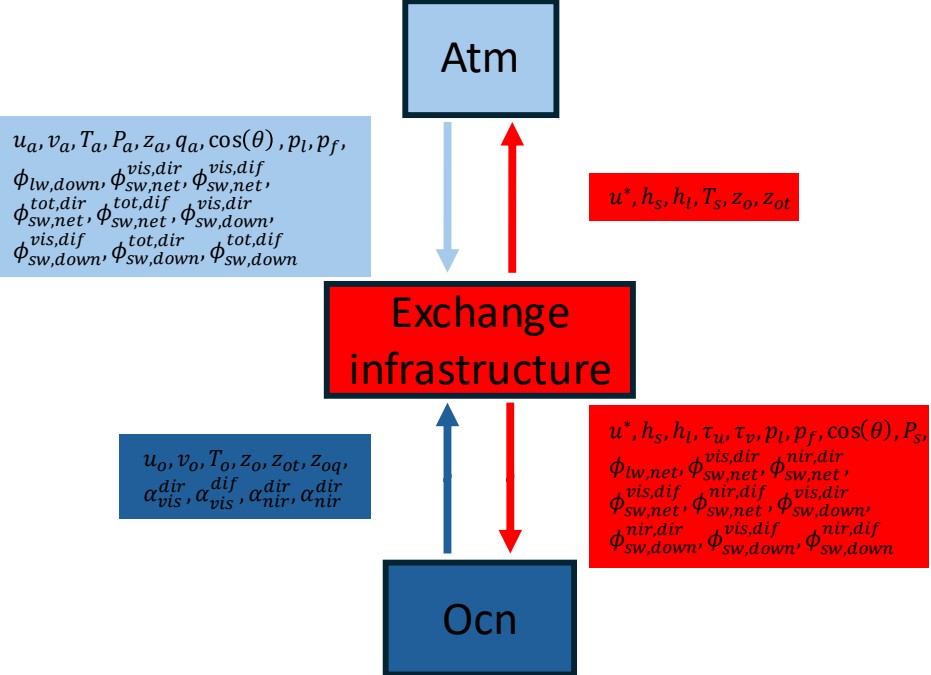

**Figure 1.** Key variables exchanges through the atmosphere, exchange grid and ocean components. Full description is shown in tables 1 and 2.

Table 1 details the atmosphere and ocean related variables projected onto the exchange grid. Atmosphere variables are categorized into dynamic and physics variables. The dynamic variables from FV3 include wind components at the lowest level, surface pressure, temperature, sea level pressure, lower layer height and tracer data. The physics variables, which reflect surface-level outputs from the model's physics suite, encompass different shortwave and longwave radiation fluxes, liquid and frozen precipitation rates, and the cosine of the zenith angle. The ocean related variables include surface parameters such as surface current components, sea surface temperature, and various roughness factors for momentum, heat, and moisture. In addition, albedos and their directional fluxes in both visible and near-infrared wavelengths are passed through the coupler, which are crucial for the energy balance between the ocean surface and the atmosphere.





Table 1: Summary of the key atmosphere and ocean variables projected onto the Xgrid. Atmosphere variables are categorized into dynamic variables (output from the FV3 dynamical core) and physics variables (surface-level outputs from the physics suite).

| Symbol | Code Variable | Meaning | Units |
|---|---|---|---|
| **Variables from FV3 $\Rightarrow$ Xgrid** | | | |
| $u_a$ | `u_bot` | Lowest level wind - zonal component | $m/s$ |
| $v_a$ | `v_bot` | Lowest level wind - meridional component | $m/s$ |
| $T_a$ | `t_bot` | Lowest level temperature | $K$ |
| $P_a$ | `p_bot` | Lowest level pressure | $Pa$ |
| $q_a$ | `tr_bot` | Lowest level specific humidity | - |
| $z_a$ | `z_bot` | Height for the lowest layer | $m$ |
| **Surface Variables from SHiELD Physics $\Rightarrow$ Xgrid** | | | |
| $\cos(\theta)$ | `coszen` | Cosine of the zenith angle | - |
| $p_l$ | `lprec` | Liquid precipitation rate | $kg/m^2 s$ |
| $p_f$ | `fprec` | Frozen precipitation rate | $kg/m^2 s$ |
| $\phi_{lw,down}$ | `flux_lw` | Downward longwave flux | $W/m^2$ |
| $\phi_{sw,net}^{vis,dir}$ | `flux_sw_vis_dir` | Net (upward - downward) direct visible shortwave flux | $W/m^2$ |
| $\phi_{sw,net}^{vis,dif}$ | `flux_sw_vis_dif` | Net (upward - downward) diffused visible shortwave flux | $W/m^2$ |
| $\phi_{sw,net}^{tot,dir}$ | `flux_sw_dir` | Net (upward - downward) direct total shortwave flux | $W/m^2$ |
| $\phi_{sw,net}^{tot,dif}$ | `flux_sw_dif` | Net (upward - downward) diffused total shortwave flux | $W/m^2$ |
| $\phi_{sw,down}^{vis,dir}$ | `flux_sw_down_vis_dir` | Downward direct visible shortwave flux | $W/m^2$ |
| $\phi_{sw,down}^{vis,dif}$ | `flux_sw_down_vis_dif` | Downward diffused visible shortwave flux | $W/m^2$ |
| $\phi_{sw,down}^{tot,dir}$ | `flux_sw_down_total_dir` | Downward direct total shortwave flux | $W/m^2$ |
| $\phi_{sw,down}^{tot,dif}$ | `flux_sw_down_total_dif` | Downward diffused total shortwave flux | $W/m^2$ |
| **Ocean Variables $\Rightarrow$ Xgrid** | | | |
| $u_o$ | `u_surf` | Surface current - zonal component | $m/s$ |
| $v_o$ | `v_surf` | Surface current - meridional component | $m/s$ |
| $T_o$ | `t_surf` | Sea surface temperature | $K$ |
| $z_0$ | `rough_mom` | Roughness length for momentum | $m$ |
| $z_{0t}$ | `rough_heat` | Roughness length for heat | $m$ |
| $z_{0q}$ | `rough_moist` | Roughness length for moisture | $m$ |
| $\alpha_{vis}^{dir}$ | `albedo_vis_dir` | Albedo for direct visible shortwave flux | - |



| Symbol | Code Variable | Meaning | Units |
|---|---|---|---|
| $\alpha_{vis}^{dif}$ | albedo_vis_dif | Albedo for diffused visible shortwave flux | - |
| $\alpha_{nir}^{dir}$ | albedo_nir_dir | Albedo for direct near-infrared shortwave flux | - |
| $\alpha_{nir}^{dif}$ | albedo_nir_dif | Albedo for diffused near-infrared shortwave flux | - |

Table 2: Summary of key Xgrid variables projected back to the atmosphere and ocean.

| Scientific Symbol | Code Variable | Meaning | Units |
|---|---|---|---|
| **Xgrid $\Rightarrow$ Atmosphere and Ocean** | | | |
| $u^*$ | ex_u_star | Friction velocity | $m/s$ |
| $h_s$ | ex_flux_t | Sensible heat flux | $W/m^2$ |
| $h_l$ | ex_flux_tr | Latent heat flux | $W/m^2$ |
| **Xgrid $\Rightarrow$ Atmosphere** | | | |
| $T_s$ | ex_t_surf | Sea surface temperature | $K$ |
| $z_0$ | ex_rough_mom | Roughness length for momentum | $m$ |
| $z_{0t}$ | ex_rough_heat | Roughness length for heat | $m$ |
| **Xgrid $\Rightarrow$ Ocean** | | | |
| $\tau_u$ | ex_flux_u | Zonal momentum flux | $N/m^2$ |
| $\tau_v$ | ex_flux_v | Meridional momentum flux | $N/m^2$ |
| $\cos(\theta)$ | ex_coszen | Cosine of the solar zenith angle | $-$ |
| $P_{sfc}$ | ex_slp | Sea level (surface) pressure | $mbar$ |
| $p_l$ | ex_lprec | Liquid precipitation rate | $kg/m^2/s$ |
| $p_f$ | ex_fprec | Frozen precipitation rate | $kg/m^2/s$ |
| $\phi_{lw,net}$ | ex_flux_lw | Net (up - down) longwave radiation flux | $W/m^2$ |
| $\phi_{sw,net}^{vis,dir}$ | ex_flux_sw_vis_dir | Net (up - down) direct visible shortwave flux | $W/m^2$ |
| $\phi_{sw,net}^{nir,dir}$ | ex_flux_sw_dir | Net (up - down) direct near-infrared shortwave flux | $W/m^2$ |
| $\phi_{sw,net}^{vis,dif}$ | ex_flux_sw_vis_dif | Net (up - down) diffuse visible shortwave flux | $W/m^2$ |
| $\phi_{sw,net}^{nir,dif}$ | ex_flux_sw_dif | Net (up - down) diffuse near-infrared shortwave flux | $W/m^2$ |
| $\phi_{sw,down}^{vis,dir}$ | ex_flux_sw_down_vis_dir | Downward direct visible shortwave flux | $W/m^2$ |
| $\phi_{sw,down}^{nir,dir}$ | ex_flux_sw_down_total_dir | Downward direct near-infrared shortwave flux | $W/m^2$ |
| $\phi_{sw,down}^{vis,dif}$ | ex_flux_sw_down_vis_dif | Downward diffuse visible shortwave flux | $W/m^2$ |
| $\phi_{sw,down}^{nir,dif}$ | ex_flux_sw_down_total_dif | Downward diffuse near-infrared shortwave flux | $W/m^2$ |





Table 2 summarizes the variables computed on the exchange grid and projected back to the atmosphere and ocean, respectively. For the atmosphere, these includes frictional velocity and ocean surface temperature, sensible and latent heat fluxes and

roughness lenghts for momentum, heat and moisture. For the ocean, sensible and latent heat fluxes, friction velocity, zonal and meridional momentums, precipitation rates and several shortwave and longwave fluxes are considered.

## 3.2    Software framework

The SHiELD_build system was initially developed by the Modeling Systems Division (MSD) at GFDL in 2015. It was then known as the fv3GFS_build system before being renamed to SHiELD_build in 2020. It employed a simplified version of the

FMS coupler, which is used by other GFDL models such as SPEAR, CM4 and ESM4. The official repository on GitHub has been actively maintained and today it supports various model workflows, including: a SOLO core FV3, SHiELD, SHiELD employing the full coupler, SHiELD and MOM6, SHiELD MOM6 and WaveWatch III (under current development). The system offers multiple compilation modes and compiler options based on model configurations, including Intel, GNU, and NVHPC compilers. The main workflow is to compile model components into libraries, starting from basic underlying infrastructure

layer such as NCEP and FMS libraries to other model components such as the atmosphere and ocean then link them through the FMS coupler to get the final executable. For a code overview, please refer to appendix A.

The FMS full coupler is a fundamental infrastructure layer serving as the main program driver and coupling component models: atmosphere, ocean, ice and land. All GFDL models utilize this driver even those running an individual model component such as AMIP (Atmospheric Model Intercomparison Project) or OMIP (Ocean Model Intercomparison Project) configura-

tions. For example, in an AMIP configuration, the atmospheric model is run using observed or forced sea surface temperatures and sea ice as boundary conditions, without coupling to a physically based ocean model that integrates in time. This setup is used to assess the performance of atmospheric models and to understand how the simulated climate responds to the prescribed conditions. The FMS full coupler supports this configuration by utilizing null modules for ocean, ice and land. An illustrative schematic is shown in figure 2. FV3 dynamical core computes prognostic dynamics quantities in the atmosphere;

SHiELD_physics drives the physics tendencies from radiation, planetary boundary layer (PBL), precipitation, etc; FMS represents the libraries described in section 2.1. The other null components, including ocean_null, ice_null and land_null, represent no-op modules to satisfy the full coupler requirements. For an OMIP configuration, the null modules of the ocean and ice are replaced by the corresponding source codes of the ocean and ice, accompanied by a null module for the atmosphere.

It should be noted that, previously, SHiELD did not utilize the FMS full coupler; however, it has now been fully integrated

with other GFDL models using the complete FMS coupler instracture, as detailed in Mouallem (2024).

In the coupled SHiELD and MOM6/SIS2, the ocean and ice null components are replaced by the MOM6 and SIS2 source codes, fulfilling the coupler requirements for the ocean and ice modules. This process is illustrated in the center schematic of figure 2. To achieve a consistent two-way coupling between the atmosphere and ocean, dynamic variables from FV3 and physics variables from SHiELD's physics must be accurately passed from the atmosphere to the exchange grid; meanwhile,

ocean variables projected onto the exchange grid must be properly passed into the atmosphere dynamics and physics suite.





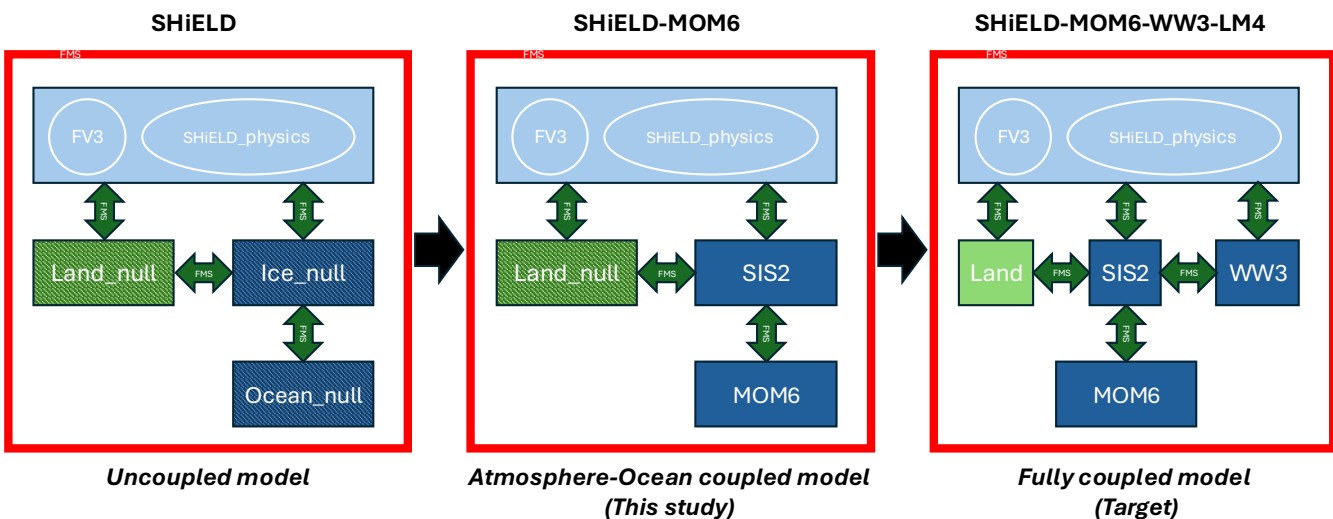

**Figure 2.** Schematic of SHiELD model component infrastructure, showing the configuration utilizing the FMS full coupler (left), the updated configuration incorporating MOM6 and SIS2 for the ocean and ice components (center) and the future target model including land and wave model components (right). Red components represent the FMS infrastructure layer, diagonal striped boxes represent null components.

The ultimate goal is to develop a fully coupled model, including comprehensive components of the land and wave models, extending the set-up presented in this paper.

It is important to note that the ice model, represented by SIS2 here, is required for the full coupler to enable complete ocean-atmosphere coupling, as atmospheric fluxes projected onto the exchange grid must first pass through the ice layer before reaching the ocean component, and vice versa. Consequently, from an infrastructure standpoint, SIS2 is included in this setup even though no ice is present in the simulations shown later on. Additionally, achieving a fully realistic configuration including realistic ice will require further development.





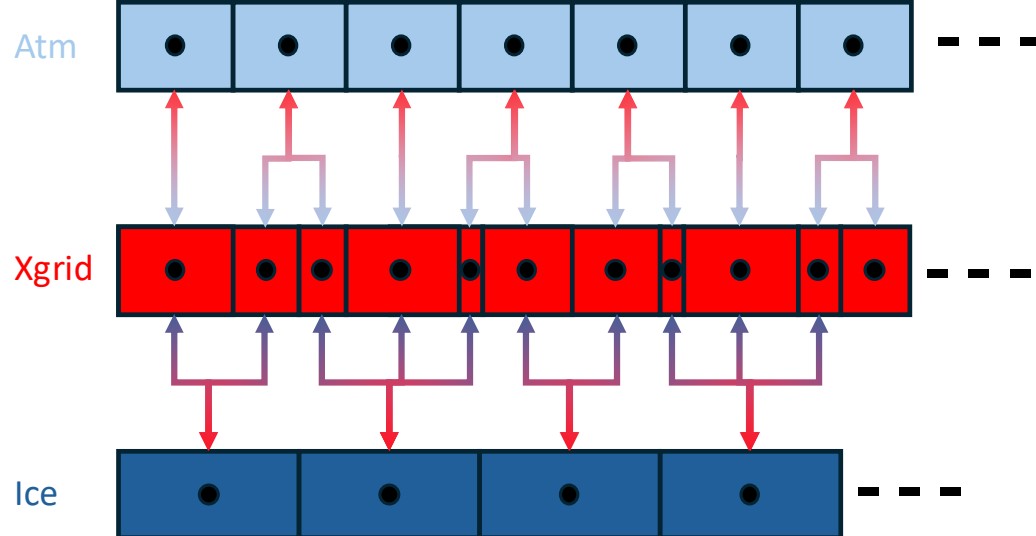

**Figure 3.** Schematic of a one-dimensional exchange grid and communication map between the atmosphere and ice components at different resolutions. The red side of the arrow indicates the step where variables are projected onto the exchange grid. The light and dark blue sections of the arrow represent the projection of variables back from the exchange grid to the atmosphere or ice components, respectively.

The GFDL exchange grid has been a main component of FMS used to facilitate data exchange between different model components, obeying scientific principles and maximizing computing efficiency (Balaji et al., 2006). Each component is dis-
cretized in a different way on a different grid depending on the science and computational requirements; for example, FV3 employs a cubed sphere grid, the ice and ocean utilize a tripolar grid. Figure 3 presents a schematic representation of the exchange process between the atmosphere and ice. The process begins with projecting relevant variables onto the exchange grid: lower-layer atmospheric variables and surface-layer ocean variables, as indicated by the light/dark blue sides of the double arrows for the atmosphere and ice, respectively. Next, fluxes and physical processes within the surface boundary layer are
computed on this intermediate grid. Finally, the updated quantities are mapped back to their respective native grids, with the projection directions represented by the red side of the double arrow. This coupling framework is highly flexible, allowing flux exchanges to occur at a user-defined timestep which should be a multiple of the model component timesteps. Notably, the exchange process maintains conservation properties, making it suitable for applications ranging from short-term weather forecasting to long-term climate prediction.





# 4 Simulations

## 4.1 Idealized doubly-periodic simulations

Table 3: Acronyms and descriptions for the three simulations: stand alone SHiELD, SHiELD coupled with MOM6, and SHiELD + MOM6 restarted from stand alone SHiELD on day 3

| Acronym | Simulation Description |
| --- | --- |
| S | SHiELD |
| SM | SHiELD + MOM6 |
| SMF | SHiELD + MOM6; Ocean coupling enabled at Day 3 |

We perform idealized simulations of an axisymmetric hurricane following the initial condition of Reed and Jablonowski (2012). No steering flow is prescribed and a constant f-plane is imposed over the whole domain. The computational domain is similar to that Gao et al. (2024) employing a square doubly periodic domain of 1000km x 1000km and a resolution of 2km centered at 20N. Different from Gao et al. (2024), we do not employ telescopic nesting and only consider the top parent grid as the computational domain. The initial vortex has a maximum wind of 20 m/s at 125 km radius. The physical parameterization is consistent with the nested domain of the T-SHiELD configuration as in Gao et al. (2021, 2023). We use the GFDL single-moment five-category microphysics scheme following Zhou et al. (2022), a turbulent kinetic energy-based eddy diffusivity mass flux (TKE-EDMF) boundary layer scheme as Han and Bretherton (2019), the Rapid Radiative Transfer Model for General Circulation Models radiation scheme described in Iacono et al. (2008) and the scale-aware deep and shallow convection parameterizations in Han et al. (2017). For simplicity, and to validate the coupling workflow, we employ a matching ocean grid in terms of domain size and resolution. The ocean model configuration is as described in section 2.3

First, we run the standalone SHiELD model for 9 days, with the results denoted as **S** in Table 3. The second case, **S**, couples SHiELD with MOM6 throughout the simulation. In the third case, we initially run the standalone SHiELD model for three days to spin up the tropical cyclone (TC) close to the rapid intensification phase, then introduce a dynamic MOM6 ocean just as the storm approaches full intensity. This case is referred to as **SMF**.







**Figure 4.** Time series of simulated maximum surface wind speed and minimum sea level pressure for the simulation **S**, **SM**, **SMF** listed in table 3.

Figure 4 shows the time evolution of the maximum surface wind speed and minimum sea level pressure for the simulations listed in Table 3. As observed, there is an initial transient period of approximately three days, after which the hurricane intensifies, reaching its peak just before day five, as indicated by the **S** curve. In the **SM** simulation, the presence of a dynamic
215  ocean facilitates energy transfer from the atmosphere to the ocean, leading to a weaker storm compared to the **S** case, in which, the prescribed constant sea surface temperature and a frictionless ocean continuously supply heat to the atmosphere, sustaining greater storm intensity. Case **SMF** initiates from case **S** at day three and slowly converges to case **SMF** just before the sixth day. This demonstrates the progressive adjustment of the coupled system, highlighting the role of air-sea interactions in regulating storm intensity and further validating the atmosphere-ocean coupling mechanism as the system evolves toward a dynamically
220  consistent state dictated by energy exchanges between the hurricane and the ocean.







**Figure 5.** 2D time evolution snapshots of surface winds and sea surface currents for simulation **SMF**. Black arrows correspond to localized velocity vectors.

Figure 5 presents the time evolution of surface winds (top two rows) and sea surface currents (bottom two rows) from days 3 to 10 for the **SMF** simulation. The atmospheric response exhibits an intensifying hurricane, with maximum surface wind speeds peaking between day 4 and day 5, followed by a gradual weakening. The wind field structure maintains a well-defined circulation throughout the simulation, with a distinct eye forming during peak intensity. In the ocean response, strong surface currents develop in conjunction with the atmospheric forcing, with peak currents observed starting the seventh day. The currents



exhibit a cyclonic structure, intensifying in response to the storm's wind stress and progressively evolving as the system reaches a more dynamically coupled state. The emergence of asymmetries in the ocean currents after day 7 highlights the increasing role of oceanic processes such as eddy formation and energy dissipation. As expected due to the Coriolis effect, these currents are deflected to the right resulting in a net outward flow away from the center. This outward transport induces upwelling, 230 bringing colder water from deeper layers to the surface, leading to a cooling effect observable in the temperature profile shown next. This figure underscores the strong two-way interaction between the atmosphere and ocean, where momentum and energy exchange drive the evolution of both the storm and the oceanic circulation.

**Figure 6.** 2D snapshots of sensible and latent heat fluxes and sea surface temperature at $t = 10hrs$ for simulations **S** (top row) and **SMF** (bottom row).

Figure 6 shows 2D snapshots of sensible and latent heat fluxes and sea surface temperature (SST) at $t = 10hrs$ for simulations **S** and **SMF**. The top row corresponds to **S**, where the SST is held constant throughout the simulation, while the bottom 235 row represents **SMF**, which includes ocean feedback mechanisms. Both the sensible and latent heat fluxes exhibit intense magnitudes in **S** compared to **SMF**. This difference is primarily attributed to the prescribed constant SST in **S**, which maintains



a sustained and continuous supply of heat to the atmosphere. In contrast, **SMF** shows a reduction in both fluxes due to SST cooling induced by oceanic upwelling, which brings colder subsurface water to the surface. The cooling effect is clear in the SST panel for **SMF**, where a well-defined cold wake forms beneath the storm. This behavior is discussed in the subsequent figures.





**Figure 7.** Time evolution of sea surface height (top), sea surface current speed (middle), and sea surface temperature (bottom) along the latitude section passed by the storm center in the **SMF** domain (left) from Day 3 to Day 9. Corresponding 2D spatial snapshots of these variables on Day 6 are shown on the right.





Figure 7 illustrates the ocean response to the hurricane from days 3 to 9, showing the daily evolution of key surface variables: sea surface height (SSH), current speed, and sea surface temperature (SST) in a latitudinal cross section at the storm center. The left panels depict the temporal evolution of these variables on each day, while the right panels show spatial snapshots on day 6. The SSH (top panel) exhibits a pronounced depression at the storm's center, which deepens over time in response to

the intensifying low pressure, mirroring the hurricane's intensification phase. The middle panel shows the evolution of surface current speed, with peak currents forming near the storm's core and intensifying through day 9. The bottom panel illustrates the SST response, where significant cooling is observed beneath the storm, driven by wind-induced mixing and upwelling of colder subsurface water. The strong correlation between SSH, current speed, and SST highlights the dynamic coupling between the ocean and the atmosphere, reinforcing the role of ocean feedback in modulating storm intensity.

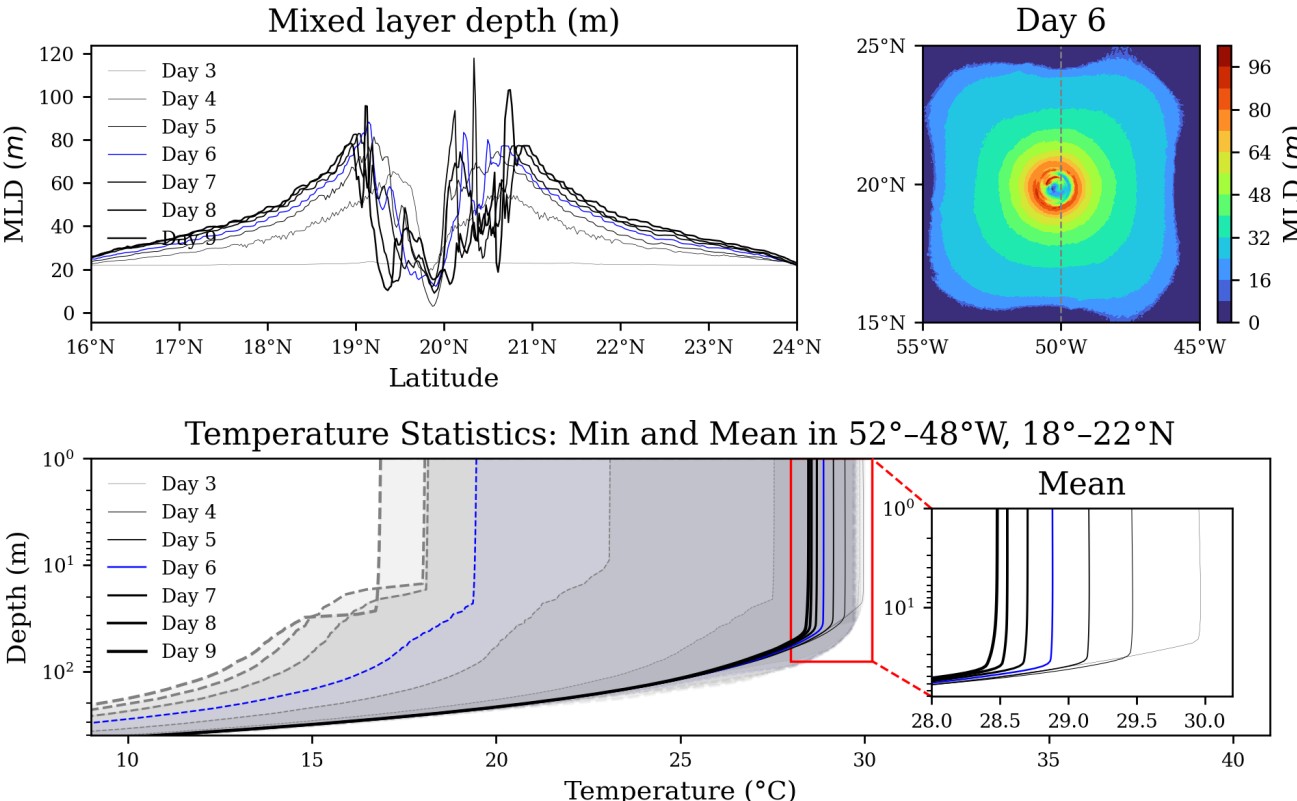

**Figure 8.** (Top) Time evolution of the Mixed Layer Depth along a central latitude cross section from Day 3 to Day 9 with a 2D spatial snapshot on Day 6 shown on the right. (Bottom) Ocean temperature minimum and mean as a function of depth from Day 3 to Day 9. Solid and dashed lines represent mean and minimum values, respectively.

Figure 8 illustrates the evolution of key subsurface quantities, the ocean mixed layer depth (MLD) and temperature in response to the simulated hurricane. We notice a progressive deepening of the mixed layer as the storm insentifies. The most pronounced deepening occurs at the storm's core around days 8 and 9. This indicates strong vertical mixing induced by





hurricane-driven wind stress and turbulent processes. The top-right panel provides a spatial snapshot of MLD on day 6, revealing a well-defined radial structure with the deepest mixing concentrated near the storm center.

The bottom panel represents the subsurface ocean temperature minimum and mean as a function of depth and time for a 4°box section at the domain center. A progressive subsurface cooling is evident for both quantities, with the thermocline deepening over time as mixing entrains colder subsurface water. This further explains the evolution of ocean surface cooling trend seen in the previous figures.

## 4.2 Realistic Simulations

In this section, we consider a realistic domain spanning the North Atlantic region. We perform a simulation of hurricane Helene to analyze and capture the ocean response. In the current simulation, the model domain is initialized at 00Z on September 26, 2024, when Hurricane Helene was still a Category 1 hurricane before undergoing rapid intensification to a Category 4 storm. The atmospheric initial conditions are taken from GFS analysis, while the ocean is initialized at rest with a constant sea surface temperature (SST). The atmosphere component runs at 1km resolution, while the ocean component runs at 3km resolution to

further validate the coupling process. The land surface model employed is NOAH-MP, which serves as the default land model in SHiELD. It is worth noting again that the coupling to the land model here is done through the SHiELD_physics suite and not at the full coupler level.

The goal of this analysis is to assess the coupling framework's performance qualitatively rather than to provide an accurate forecast of Helene's track and intensity. By using a quiescent ocean with constant SST, we can isolate the atmospheric effects

on the ocean and evaluate the coupled model's behavior without additional environmental complexities.



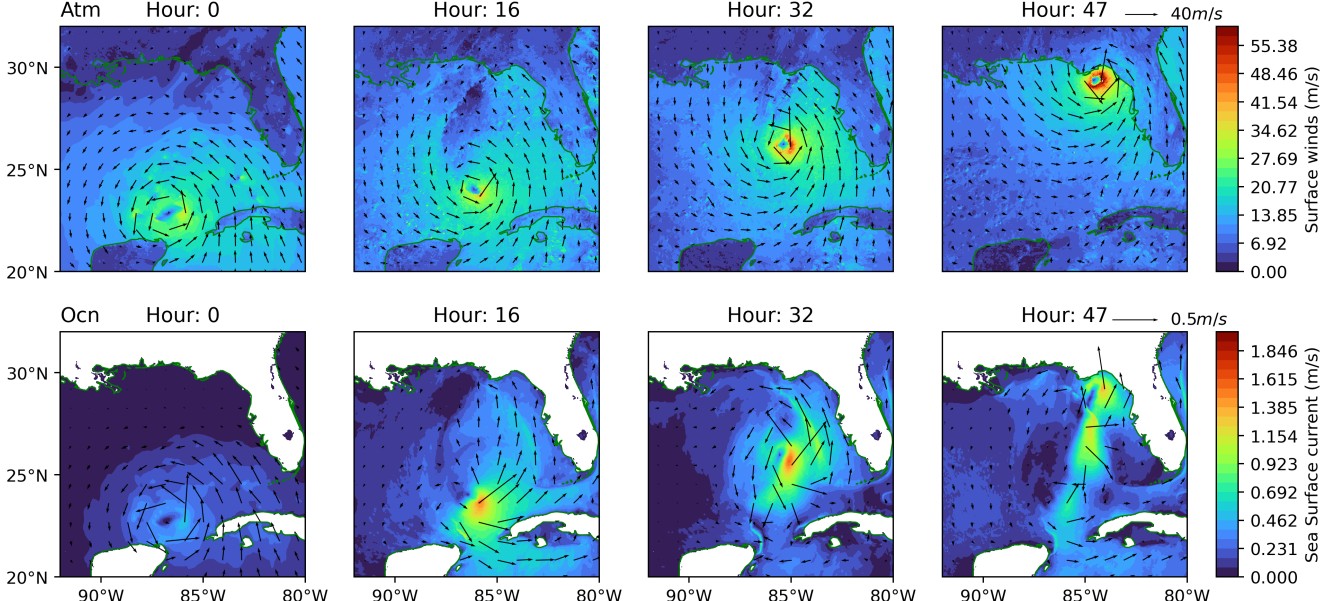

**Figure 9.** 2D time evolution snapshots of surface winds and sea surface currents for hurricane Helene. Black arrows correspond to localized velocity vectors.

Figure 9 illustrates the interaction between atmospheric surface winds (top row) and ocean surface currents (bottom row). Initially, a symmetric wind field surrounds the storm, but as the hurricane intensifies, wind asymmetries emerge, particularly near the coastline. The strong hurricane surface winds generate rapid responses in the ocean, leading to the formation of strong currents. The ocean surface currents display a classic hurricane-induced structure Bender et al. (1993), with intensified flow on the right-hand side of the storm track due to inertial resonance Price (1981). As the storm moves northward, ocean currents become more pronounced along the continental shelf, highlighting the role of bathymetric effects in modulating the ocean response.





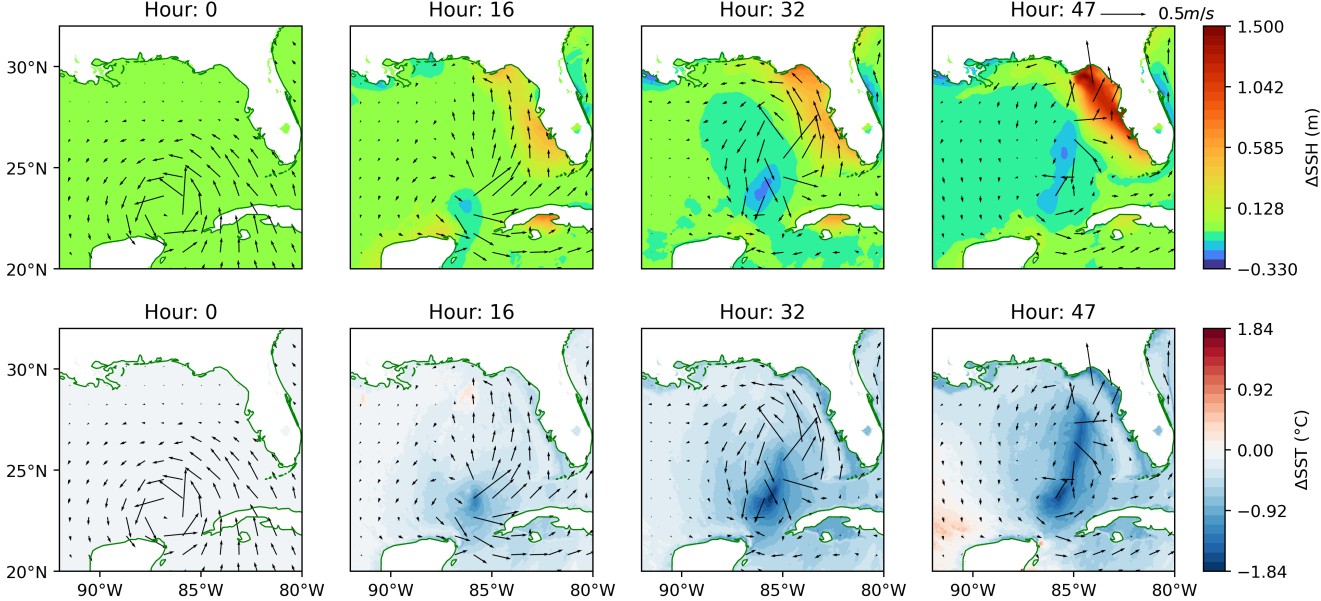

**Figure 10.** 2D time evolution snapshots of sea surface height (top) and temperature (bottom) compared to their initial values for hurricane Helene. Black arrows correspond to localized velocity vectors.

Figure 10 presents the evolution of SSH and SST in response to the hurricane in 16hours intervals compared to their initial values. The SSH panels (top row) indicate a developing storm surge along the northern Gulf Coast, with significant sea level anomalies forming at the storm center. Over time, the SSH field shows a pronounced dip directly under the storm, corresponding to the atmospheric pressure drop as seen in the idealized test case, while coastal regions experience positive SSH anomalies indicative of storm surge. The SST panels (bottom row) reveal strong ocean cooling along the hurricane's path, driven by wind-induced mixing and upwelling. This cooling intensifies as the storm strengthens, particularly in regions where wind stress is highest. This is also in line with the behavior seen in the idealized test case.

# 5   Model scalability

Parallel efficiency is a key factor for the performance of coupled ocean-atmosphere models, particularly when simulating large-scale, high-resolution problems. There are already reports on the scalability of standalone SHiELD and standalone MOM6. In this section, we investigate the parallel efficiency of the coupled model SHiELD-MOM6 , focusing on three main objectives: (a) validating the scalability of the full coupled model (b) demonstrating that the usage of the FMS coupler and exchange grid can effectively handle massive parallel simulations, and (c) assessing the coupling process additional computational overhead.

We evaluate the parallel speed-up of the SHiELD/MOM6 system to understand its performance under varying numbers of CPU cores, investigating strong scaling by simulating a constant size problem with different core configurations and weak





scaling by simulating increasingly larger problem sizes while maintaining a constant number of grid cells per processing element.

295    These tests were performed on the supercomputer GAEA, operated by National Climate-Computing Research Center and located at the Oak Ridge National Laboratory (ORNL). The C6 cluster partition of Gaea is a HPE-EX Cray X3000 system with 2048 compute nodes (2 x AMD EPYC 9654 2.4GHz base 96-cores per socket), HPE Slingshot Interconnect, 384GB DDR4 per node; 584TB totaling an expected peak of 11.21 PF.

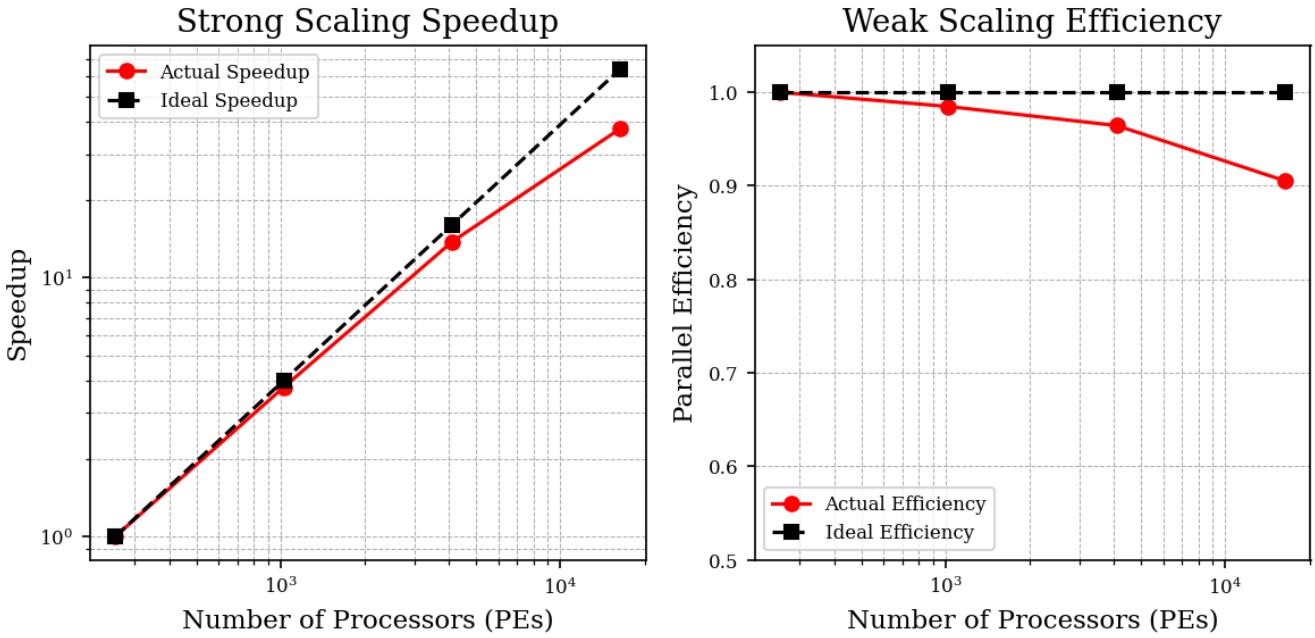

**Figure 11.** Strong scaling (left) and weak scaling (right): Actual speedup/efficiency (red circles) compared to ideal speedup/efficency (black squares) as a function of the number of PEs.

For the strong scaling, we consider the idealized doubly periodic simulations, shown in section 4, with a fixed larger domain
300   size of 1024x1024 grid cells. Scaling tests were conducted using various processor layouts: 8x16, 16x32, 32x64, 64x128 which corresponds to 128x64, 64x32, 32x16, 16x8 grid cells per processing element, respectively. Additionally, two cores were allocated per task. The strong scaling performance of the model is illustrated in figure 11, which shows the achieved speedup factor as a function of the number of Processing Elements (PEs). The actual speedup (red) is compared to the ideal linear scaling (black). The reference case considers the case with the smallest number of PEs. The results demonstrate a near ideal
305   scaling up to 16 thousands PEs, after which the actual speed up begins to deviate from the ideal case. Despite this deviation, the model maintains strong parallel efficiency, demonstrating substantial computational gains with increasing processor count. In addition to strong scaling, we also performed a weak scaling analysis where we fixed the grid cell count per PE at 32x16 and increased the numbers of PEs up to 16 thousands PEs. The results indicate that the model maintains high efficiency (>90%)





demonstrating excellent parallel scalability. The slight decrease in efficiency at higher processor counts can be attributed to increased communication overhead and load imbalance. Overall, the strong and weak scaling results demonstrate the model's excellent parallel performance, making it well-suited for large-scale simulations on high-performance computing architectures.

## 6 Conclusion and Future work

In this work, we have developed and validated the new coupled SHiELD-MOM6 model that advances our ability to simulate complex interactions between the atmosphere and ocean. Built upon the robust frameworks of GFDL's FMS, FV3 based SHiELD atmospheric model, MOM6 ocean model, and SIS2 sea ice simulator, the coupled system achieves a two-way atmosphere/ocean integration through the FMS coupler and exchange grid. During the implementation process, we have ensured that the coupling method enables precise, conservative transfer of dynamic and physics variables between the atmosphere and ocean within the essential physical processes such as momentum, heat and moisture exchanges for their accurate representation.

The idealized and realistic scenario simulations highlight the new system's capabilities. In the idealized hurricane test case, the model successfully captured key features of the storm development, including its intensification phase, evolution of storm structure, and corresponding ocean responses like surface current adjustment and wind-induced upwelling. Similarly, realistic simulations of hurricane Helene of September 2024 demonstrated the model capability to simulate and reproduce complex phenomena such as storm surge development, coastal current modulation, and significant sea surface temperature changes driven by air–sea interactions.

Additionally, scalability tests on high-performance computing systems revealed that the SHiELD-MOM6 model is not only scientifically robust but also computationally efficient for the extent to the current configuration and tests. The effective parallel performance achieved through the optimized coupling strategy and exchange grid paves the way for its application in operational settings, ranging from short-term weather forecasts to extended future climate simulations.

Overall, the new SHiELD-MOM6 model represents a major advancement in coupled model development at GFDL. Its flexible and modular design, combined with state-of-the-art numerical frameworks and infrastructure, provide a solid foundation for future studies and forecasts on severe weather systems which need correct representations of air-sea interactions like hurricanes. Current development efforts include integrating additional model components such as Wavewatch III for wave dynamics, further refinement of physical parameterizations, and extensive validation against observational datasets. These improvements will further enhance our understanding of air–sea interactions and contribute to more accurate forecasting and climate prediction efforts.

This work highlights the critical role of unified modeling approaches in addressing the inherent complexities of coupled systems and sets a foundation for continued progress in model development for weather and climate science.



**Appendix A**

The official code repositories are all on GitHub under NOAA-GFDL. The main branches are up to date and continously under
developement. The source code for the main components discussed in section 2 are below:

- https://github.com/NOAA-GFDL/GFDL_atmos_cubed_sphere

- https://github.com/NOAA-GFDL/FMScoupler

- https://github.com/NOAA-GFDL/FMS

- https://github.com/NOAA-GFDL/atmos_drivers

- https://github.com/NOAA-GFDL/SHiELD_physics

- https://github.com/NOAA-GFDL/land_null

- https://github.com/NOAA-GFDL/ice_param

- https://github.com/NOAA-GFDL/MOM6

- https://github.com/NOAA-GFDL/SIS2

The source code for the build component is under:

- https://github.com/NOAA-GFDL/SHiELD_build

*Code availability.* SHiELD-MOM6 is under active development and can be built and run from the latest official resleases from the NOAA-GFDL GitHub repository as listed in appendix A. The simulation files and source code for the static version of the model used in this study are available at https://doi.org/10.5281/zenodo.15178709 (Mouallem, 2025)

*Competing interests.* The authors declare that they have no conflict of interest.

*Disclaimer.* The statements, findings, conclusions, and recommendations are those of the author(s) and do not necessarily reflect the views of the National Oceanic and Atmospheric Administration, or the US Department of Commerce.

*Acknowledgements.* We thank Jacob Steinberg and Thomas Robinson for their review and useful comments that improved the quality of the manuscript. Mouallem, Gao and Zhang are supported under awards NA18OAR4320123, NA23OAR4320198, NA23OAR4050432I from



the National Oceanic and Atmospheric Administration, U.S. Department of Commerce. This work is also supported by the NOAA research
      Global-Nest Initiative and the Bipartisan Infrastructure Law (BIL).



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
