# Peer review of "Development of a High-Resolution Coupled SHiELD-MOM6 Model. Part I – Model Overview, Coupling Technique, and Validation in a Regional Setup"

_EGUsphere, 2025_

## Author Response (AR1)

*We would like to thank the referee for the careful consideration of our work; we also appreciate the useful comments and suggestions to help us improve the quality of our contribution. We have provided a response to the specific comments below and referred to the subsections of the corresponding changes in the new version of our manuscript. Please note that the modifications to the manuscript are written in blue; the new additions to the manuscript are in red.*

**Referee 1**

**Summary**

This paper describes the coupling infrastructure for the model SHIELD/MOM6 and clearly demonstrates the effects of the coupling by running three different configurations: an uncoupled atmospheric case, an atmosphere/ocean coupled case, and a mixed case in which the uncoupled SHIELD is run for three days and the ocean coupling in turned on for the rest of the simulation. Two test cases are run: an idealized tropical cyclone (TC) and a realistic hurricane: Hurricane Helene 2024. In both test cases, atmospheric fields such as sea surface pressure and wind velocity, and ocean fields such as sea surface temperature and sea surface currents responded as expected to the energy and momentum exchange between the ocean and the atmosphere, demonstrating the success of the coupling infrastructure.

The paper is well written, the objectives are clearly stated and the results support their conclusions. I consider this work relevant because the SHIELD/MOM6 model would add to the diversity of the air-sea couple models, in particular to the regional hurricane forecasting models. The diversity this model could add comes from a different couple infrastructure, different atmospheric physics and different horizontal and vertical grid resolution.

*RESPONSE: We would like to thank the referee for their helpful and supportive comments about our work. We have carefully addressed all of the fundamental questions and concerns raised below, with corresponding revisions made throughout the manuscript.*

**Individual scientific questions/issues:**

**1. Line 18 can you explain why it is required that the atm fluxes have to pass first through the ice layer and then to the ocean component? Does this have an extra computational cost?**

*RESPONSE: In the GFDL coupled model framework, the atmosphere communicates with the land and sea ice components via the exchange grid. It is worth noting that the ice and ocean run on the same grid, eliminating the need for interpolation when exchanging fluxes between them. From a physical standpoint, when sea ice is present, it acts as a barrier between the atmosphere and ocean (modulating fluxes), Therefore, atmospheric fluxes have to pass through this layer before being passed to the ocean and vice-versa. This ensures realistic representation of air–ice–ocean interactions. To keep the algorithm simple and consistent, the flux route structure is used even over open waters; however, in this case there is no physical atmosphere-ice and ice-ocean interactions. This approach keep the algorithm consistent without introducing computational cost. It is worth noting that the timescales of the physical ice–ocean interactions differ from those of the exchange grid in practice.*
*We added additional text in section 3.2 to clarify this point.*

**Technical corrections:**

**1. Line 52: Define the HPC acronym**

*RESPONSE: Thanks for pointing this out, the acronym is now defined in the text.*

**2. Line 58: "The development of new component models is also outside its direct scope but supported by infrastructural changes within FMS. The collaborative software review process for contributed models is crucial to FMS": These sentences are difficult to interpret. Please remove or rewrite clearly**

*RESPONSE: Thanks for pointing this out, this sentence has been revised.*

**3. Line 90: a modified version of the Mixed 90 Layer Ocean of Pollard et al. (1973)? Why is this needed if the atmospheric and ocean components are fully coupled?**

*RESPONSE: This description is intended to describe the standalone current SHiELD configurations as described in the references in that section: (Harris et al., 2020; Zhou et al., 2024; Gao et al., 2021, 2023; Harris et al., 2019; Kaltenbaugh et al., 2022)*

**4. What do you mean by telescoping z\* vertical coordinate?**

*RESPONSE: We have added a reference to a figure in another publication that demonstrates the telescoping resolution.*

**5. Line 117: "The surface grid spacing is is 2m in this configuration". Do you mean the vertical grid spacing? "is " is written twice**

*RESPONSE: Thanks for noticing this, the text has been updated.*

**6. How is u\* calculated in the Xgrid?**
*RESPONSE: We have added a paragraph to offer a high-level overview of how the momentum and heat fluxes are calculated in FMS coupler.*

**7. Is To (sea surface temperature) the temperature at the shallowest MOM6 depth level?**

*RESPONSE: This is true, we added text to clarify this point.*

**8. How is Ts (sea surface temperature) obtained in the Xgrid?**

*RESPONSE: Sea surface temperature is projected from the ocean model to the exchange grid. This projection involves an interpolation from the model parent grid to each corresponding cell on the exchange grid, as described in section 3.2. Additional information can be found in Balaji 2006 as mentioned in that section.*

**9. Line 200: The radius of maximum wind (RMW) of 125 km for the initial vortex is unrealistically large. The RMW is typically around 20-50 km. Why did you choose 125 km as RMW?**

*RESPONSE: The reviewer is correct that for a mature hurricane, the radius of maximum wind (RMW) typically ranges from 20 to 50 km. However, in our idealized simulation, the model is initialized with a relatively weak vortex, and an RMW of 125 km for a weak tropical storm is not unusually large.*

**10. Fig. 5 shows results from the SMF. In this configuration the coupling is introduced on day 3. Why do the sea surface currents seem to be well developed on day 3?**

*RESPONSE: Thanks for noticing that. The text and caption has been corrected to reflect the exact time of the first figure for the ocean and atmosphere which is 1hr into day 3.*

**11. Line 200, 238: "Leading to a cooling effect observable..." You should also mention that not only upwelling causes the SST cooling, but in realistic cases, most of the cooling is caused by the wind-induced vertical mixing at the base of the ocean surface boundary layer**

*RESPONSE: Thanks for the suggestion, we have modified the text accordingly.*

**12. Fig. 6 For the SMF case, why at t=10h there is coupling with the ocean? I understood the coupling for this case started at day 3**

*RESPONSE: Thanks for noticing that. The text and caption has been corrected to reflect the exact time which is t=10hrs into day 3.*

**13. Fig. 7 and 9 left panels: please use different line styles or colors for days 7, 8 and 9**

*RESPONSE: Thanks for the suggestion. The line styles corresponding to day 7,8,9 have been updated for better clarity.*

**14. Fig. 8 bottom figure: This figure is a bit confusing. It would be clearer if the minimum (with the shades) and the mean values are presented in different panels.**

*RESPONSE: Thanks for the suggestion. The minimum and mean values are now presented in different panels for better clarity.*

**15. Fig. 9: The ocean currents at t=0 hours are different from zero. Why? It was mentioned that the ocean was initialized at rest**

*RESPONSE: Thanks for noticing that. The ocean is indeed initialized at rest. The first column actually corresponds to t=1hr. The figure has been revised.*

*We would like to thank the referee for the careful consideration of our work; we also appreciate the useful comments and suggestions to help us improve the quality of our contribution. We have provided a response to the specific comments below and referred to the subsections of the corresponding changes in the new version of our manuscript. Please note that the modifications to the manuscript are written in blue; the new additions to the manuscript are in red.*

**Referee 2**

**General comments**

The manuscript introduces a recent developed high-resolution coupled atmosphere–ocean model, integrating GFDL's SHiELD (FV3-based atmospheric model), MOM6 ocean model, and SIS2 sea ice simulator via the FMS coupler with an exchange grid. The coupled model was used for some idealized hurricane case simulations to demonstrate the capability and impact of ocean coupling on hurricane simulation. Computation scalability was also explored for this coupled modeling system. Although there are already other coupled global and regional forecast systems (available in various coarse and high resolutions, in both research and operational applications), development and enabling the high-resolution regional coupled SHiELD-MOM6 system definitely expands the GFDL's model suites/capabilities, as well as contributes to the diversity of multiple high resolution regional coupled modeling systems.

Overall, the manuscript is well prepared. However, there are a few minor concerns/comments, which I think need to be addressed and/or clarified, before being fully accepted for publication.

*RESPONSE: We would like to thank the referee for their helpful and supportive comments about our work. We have carefully addressed all of the fundamental questions and concerns raised below, with corresponding revisions made throughout the manuscript.*

**Specific comments**

1- Page 10, last paragraph (lines 178-182): It is mentioned that, even though sea ice component/coupling is not involved/needed for the simulations related to this manuscript's case studies, the SIS2 component is still included/configured to facilitate the FMS and Xgrid related atmosphere-ocean coupling. A related concern/question/clarification: is there any additional overhead (additional computation/communication cost) by involving a not-actually-used component (SIS2) in this specific atmosphere-ocean (SHiELD-MOM6) coupled model? Meanwhile, for schematics in Figure 2 (both middle and right panels), it seems to me the communication between FV3 atmosphere and MOM6 ocean components (as well as between WW3 wave and MOM6 ocean components) all need go through the SIS2 component (through FMS). However, FV3 and WW3 (FV3 and Land) in the right panel will be able to communicate directly through FMS. Wondering if these are designed/implemented with any specific considerations/reasons).

*RESPONSE: In the GFDL coupled model framework, the atmosphere communicates with the land and sea ice components via the exchange grid. It is worth noting that the ice and ocean run on the same grid, eliminating the need for interpolation when exchanging fluxes between them. From a physical standpoint, when sea ice is present, it acts as a barrier between the atmosphere and ocean (modulating fluxes), Therefore, atmospheric fluxes have to pass through this layer before being passed to the ocean and vice-versa. This ensures realistic representation of air–ice–ocean interactions. To keep the algorithm simple and consistent, the flux route structure is used even over open waters; however, in this case there is no physical atmosphere-ice and ice-ocean interactions. This approach keep the algorithm consistent without introducing computational*

*cost. It is worth noting that the timescales of the physical ice–ocean interactions differ from those of the exchange grid in practice.*

*We added additional text in section 3.2 to clarify this point.*

*The wave model implementation is still work in progress. At the moment, it is separated from the ice-ocean coupling and maintained as a separate component so it can be more flexible, e.g., it can be run in atmosphere-wave mode with no ocean model or it can be run on a different grid from the ocean model.*

**2- Section 4.2, in this section, a real hurricane case (Helene 09L2024) was chosen as a case study in this manuscript. However, it looks to me, even though the atmospheric model component was initialized from GFS analysis, the ocean component was initialized by using idealized ocean conditions (constant SST with prescribed vertical structure, and with a rest ocean with no initial currents). In this case, I would suggest either to choose initialize the MOM6 ocean component with a more realistic initial condition (e.g. from RTOFS, or other available analysis/forecast), or at least change the title of this section into something like "Realistic Hurricane Simulation with Idealized Ocean Condition" or something similar. Related descriptions and mentioning in other locations of the manuscript should also be updated accordingly.**

*RESPONSE: We added an additional plot (figure 11) showing a hurricane Helene simulation with an ocean initialized from realistic initial condition to extend this analysis.*

**3- In Section 5, regarding the scalability testing, it's not clear to me how the load balance were considered/achieved between the different coupling components (FV3 and MOM6). For example, how many PEs were used by FV3 and MOM6, respectively? Meanwhile, how frequently does the two components exchange variables between each other (coupling time step)? Another related question, did you compare/assess the coupling overhead, saying wall clock times comparing coupled runs against uncoupled atmospheric and oceanic runs?**

*RESPONSE: Thanks for pointing this out, we revised the text to add additional information. The atmosphere and ocean run serially on the same processors. The coupling timestep is still unchanged and corresponds to the atmosphere physics timestep (or the long atmosphere timestep). Additional work is needed to assess model scalability if the atmosphere and ocean components run concurrently on separate processor sets. This includes identifying an efficient processor distribution strategy that optimizes load balance for practical applications. We haven't performed a thorough analysis to compare coupled vs uncoupled wall clock time yet.*

**4- Lines 208-209: "The second case, S, couples SHiELD with MOM6 ...." should be "SM" instead of "S" here?**

*RESPONSE: Thanks for pointing this out. The text is now corrected.*

**5- Line 216: Please clarify what does this "a frictionless ocean" mean here?**

*RESPONSE: Thanks for pointing this out. We have revised the text.*